# Hidradenitis Suppurativa and JAK Inhibitors: A Review of the Published Literature

**DOI:** 10.3390/medicina59040801

**Published:** 2023-04-20

**Authors:** Fabrizio Martora, Massimiliano Scalvenzi, Angelo Ruggiero, Luca Potestio, Teresa Battista, Matteo Megna

**Affiliations:** Section of Dermatology—Department of Clinical Medicine and Surgery, University of Naples Federico II, 80131 Naples, Italy

**Keywords:** hidradenitis suppurativa, JAK inhibitors, treatment, adalimumab, secukinumab, guselkumab, risankizumab, ustekinumab, bimekizumab, tofacitinib, ruloxitinib, upatacitinib, pathogenesis

## Abstract

*Background*: Hidradenitis suppurativa (HS), also known as acne inversa or Verneuil’s disease, is a chronic, inflammatory, recurrent, and debilitating skin disease of the hair follicles characterized by inflammatory, painful, deep-rooted lesions in the areas of the body characterized by the presence of the apocrine glands. Unfortunately, huge unmet needs still remain for its treatment. *Objective*: The purpose of our review was collecting all cases, case series, trials, and ongoing studies available in the literature on the use of this class of drugs for HS. *Materials and Methods*: The investigated manuscripts included trials, reviews, letters to the editor, real-life studies, case series, and reports. Manuscripts were identified, screened, and extracted for the relevant data following the PRISMA (preferred reporting items for systematic reviews and meta-analyses) guidelines. *Results*: We selected 56 articles of which 25 met the selection criteria for our review. Among the JAK inhibitors to date, there is only one published clinical trial in the literature (Janus kinase 1 inhibitor INCB054707), a real-life study with 15 patients up to week 24 in which upadacitinib was used and a case series where tofacitinib was successfully used. Conversely, there are several ongoing clinical trials. *Conclusions*: Results to date in the literature show promising levels of efficacy and the safety of JAK inhibitors in HS. Several clinical trials are underway from which it will be very important to compare the available data. There are still too few studies conducted with a low sample size, so it remains critical to investigate this issue further in the future with a real-life study involving a large sample of patients in order to provide safe and viable therapeutic alternatives for HS.

## 1. Introduction

Hidradenitis suppurativa (HS), also known as acne inversa or Verneuil’s disease, is a chronic, inflammatory, recurrent, and debilitating skin disease of the hair follicles that usually presents after puberty with inflammatory, painful, deep-rooted lesions in the areas of the body characterized by the presence of the apocrine glands: the armpits, breast, groin, gluteal area, and perianal area [1,2]. HS is now considered as a pathology of the pilosebaceous follicle unit [3].

The treatment of HS has always been a real challenge for dermatologists; mild HS forms are usually treated with conservative treatment such as topical resorcinol or clindamycin or hair laser epilation while moderate to severe forms undergo long-term antibiotics or may be candidates for biological therapies [4,5,6,7,8,9,10]. To date, the only biologic drug approved for HS is adalimumab, an anti-tumor necrosis factor (TNF)-α drug [11,12,13].

However, the efficacy of adalimumab in daily practice is highly variable [14,15], and the need to identify new therapeutic targets for patients with HS still remains a significant unmet need.

To date, there are ongoing phase III or phase II studies in the literature with anti-interleukin (IL)-17 drugs bimekizumab and secukinumab [16,17,18] Particularly, secukinumab is the biologic drug in the most advanced stage of clinical development for HS, showing promising results [16]. The role of IL-23 is still very controversial; indeed, to date, the only published phase II study involves the use of risankizumab at doses of 180 mg or 360 mg, which showed no difference: primary endpoint (HiSCR) was achieved by 46.8% of patients with risankizumab 180 mg, and 43.4% with risankizumab 360 mg [19]. Data on the clinical trial involving another anti-IL23, guselkumab, have not yet been published (ClinicalTrials.gov identifier: NCT03628924).

With regard to other biologics, there is still very little evidence in the literature on the treatment of HS targeting IL 36 (spesolimab). In this case, the activation of neutrophils could favor the outcome of the treatment because it adapts to the pathogenesis of HS; the results of the spesolimab study are awaited from the phase II trial [18].

More evidence is present thanks to the use of Janus kinase (JAK) inhibitors. Inhibition of Janus JAK 1 signaling in HS has shown clinical efficacy only at the highest dosages, highlighting that careful surveillance of the balance between the safety and efficacy of JAK inhibition is warranted [19].

These drugs act on JAKs, a family of four proteins: JAK1, JAK2, JAK3, and TYK2 [20]. Thanks to the activation of intracytoplasmic transcription factors such as signal transducer and transcription activation (STAT), they manage to modulate the inflammatory process [20].

Consequently, after activation, they move into the nucleus forming dimers, positively or negatively modulating thousands of genes [21].

The administration of these drugs involves both the oral and topical route, therefore, their use in various chronic inflammatory diseases including HS was immediately of great interest to dermatologists [22,23]. The purpose of our review was to bring together all of the trials, cases, case series, and ongoing studies available in the literature on the use of this class of drugs for HS.

## 2. Materials and Methods

For the current review, a literature search was performed on the PubMed, EBSCO, Embase, Google Scholar, Cochrane Skin, and MEDLINE databases (until January 2023). Research was performed by using and matching the following terms: “hidradenitis suppurativa”, “treatment”, “jak inhibitors”, “target molecules”, “upatacitinib”, “tofacitinib”, “ruloxitinib”, “deucravacitinib”, “porvocitinib”, “real life”. Investigated manuscripts included trials, metanalyses, reviews, letters to the editor, real-life studies, case series, and reports. Manuscripts were identified, screened, and extracted for the relevant data following the PRISMA (preferred reporting items for systematic reviews and meta-analyses) guidelines [24] (Figure 1).

The bibliography was also analyzed to include articles that could have been missed. Only English language manuscripts were included. This article is based on previously conducted studies and does not contain any studies with human participants or animals performed by any of the authors.

### 2.1. Eligibility Criteria and Study Selection

Inclusion criteria were as follows: (1) placebo- or active comparator-controlled human studies; (2) trials focusing on the efficacy and safety of jak inhibitors in HS; (3) RCTs with at least one of the following outcomes reported: Hidradenitis Suppurativa Clinical Response (HiSCR), Dermatology Life Quality Index (DLQI), and adverse events. However, real-life studies were also included. Since there are still very few real-life studies on the use of these drugs, case reports and case series were selected and not excluded. The extracted data of interest contained the baseline demographics (age, gender, body mass index, and severity), the number of participants, treatment regimens, and outcome evaluation. Our primary outcome was HiSCR. Secondary outcomes were DLQI 0/1, mean DLQI change from baseline, and adverse effects. 

### 2.2. Risk of Bias Selection

The quality of enrolled studies was appraised by two investigators (F.M. and T.B) using the revised Cochrane risk-of-bias tool for RCTs [25]. Through evaluating the methodology, the overall risk of bias was judged as low, high, or some concern. If differing opinions existed, the third author (M.M.) was consulted for arbitration. 

## 3. Results

We selected 56 articles of which 25 met the selection criteria for our review.

We have divided the results section into various sub-sections according to the class of JAK inhibitor who had a positive or negative experience with the management of HS, and we added a sub-section where all ongoing clinical trials were included.

### 3.1. Janus Kinase 1 Inhibitor INCB054707

To date, there is only one phase II study in the literature where the Janus kinase 1 inhibitor INCB054707 has been investigated for HS. This study is registered on ClinicalTrials.gov (NCT03569371 and NCT03607487) [26].

The study was divided into two arms; on the one hand, patients received INCB054707 15 mg once daily (QD; Study 1) in the open for 8 weeks; then patients were randomized to INCB054707 30, 60, or 90 mg QD or the placebo (3:1 in each cohort) for 8 weeks [26]. A follow-up visit was conducted 30 days after the end of the study to confirm the safety and efficacy data.

The eligibility criteria were as follows: age between 18 and 75 years; HS of moderate-to-severe grade (Hurley disease stage II/III); lesions present in at least two anatomical sites; a total abscess and inflammatory nodule count ≥ 3. 

For both study arms, the primary endpoint was safety and tolerability including severity adverse events (AEs) and laboratory test results.

Secondary endpoints included HS clinical response (HiSCR) and other efficacy measures.

A total of 10 patients were enrolled in arm 1 and 35 patients in arm 2 [26]. The authors reported that a total of 70% of patients and 81% of patients reported a treatment-related AEs, respectively. In particular, headache, fatigue, and thrombocytopenia were the most frequent AEs for patients taking the treatment, while headache and diarrhea were the most frequent among patients receiving the placebo. The authors associated thrombocytopenia with drug dosing as only four patients developed this adverse event and all were on 90 mg; however, 90 mg treatment was temporarily discontinued for only 2 weeks; indeed, these patients re-started the therapy.

No AEs, in general, led to the definitive discontinuation of treatment [26].

As for the results of arm 1, three patients (43%) achieved HiSCR at week 8. The proportions of patients achieving an abscess and inflammatory count (AN) count of 0–2 at week 8 and the safety follow-up were 43% and 57%, respectively. 

Regarding study arm 2, 17 patients [65% in total: 30 mg (56%), 60 mg (56%), 90 mg (88%)] receiving INCB054707 vs. four patients (57%) receiving the placebo achieved HiSCR at week 8. 

Half of the patients receiving INCB054707 (30 mg, 44%; 60 mg, 44%; 90 mg, 63%) at week 8 achieved an AN count of 0–2 [26].

The improvements were dose-dependent, with the best results being particularly observed in patients on higher dosages of INCB054707 [26].

Improvements were observed in both the patients’ quality of life, lesion count, and HISCR. The real limitation of this study was the sampling, which was too low, but the authors conclude that a phase II, placebo-controlled study exploring three dose levels and including about 200 patients (NCT04476043) is ongoing and should provide further evidence of the safety and efficacy profile of INCB054707 in patients with HS [26].

### 3.2. Upadacitinib

Upadacitinib is an oral selective JAK-1 inhibitor with high selectivity for JAK1 and its signal transduction molecules. To date, it is approved and recommended by the European Medicines Agency (EMA) and the Food and Drugs Administration (FDA) for the treatment of moderate-to-severe rheumatoid arthritis (RA), especially in patients with intolerance to MTX or who have had an inadequate response to the substance, and recently, for moderate-to-severe atopic dermatitis of adult and adolescent (12 years and older) candidates to systemic therapy [27,28,29,30,31]. No phase II or III trials have been published so far for upadacitinib in the literature. However, Kozera et al. performed a real-life study testing upadacitinib in HS [32].

The authors conducted a retrospective cohort study of all patients with moderate to severe HS treated with upadacitinib monotherapy. The study design was as follows: all patients received 15 mg of upadacitinib daily up to week 4. If the clinical response was not sufficient after 4 weeks, treatment doses were increased to 30 mg daily [32]. Follow-up visits were performed at weeks 12 and 24. All patients had initial baseline screening tests: HIV, hepatitis B, hepatitis C, and liver tests, then at week 4, the complete blood count, biochemistry, lipid, and creatine kinase levels were performed. During all 24 weeks, no other treatments such as antibiotics, corticosteroids, or surgery were carried out. HISCR, IHS4, DLQI, and VAS pain scale were evaluated as endpoints.

The results were very positive; in total, 15 patients (75%) achieved HiSCR 50 at week 4, growing up to 100% at week 12 with the results being maintained up to week 24. 

HiSCR75 was achieved in six patients (30%) at week 4 and19 patients (95%) at week 12; this result was maintained up to week 24.

HiSCR90 was achieved in four patients (20%) at week 4 and increased to six patients (30%) at week 12; these results were maintained to week 24.

The authors do not provide precise data on IHS4, DLQI, and VAS pain scale, they only mention statistically significant reductions in their manuscript. Regarding the side effects, one patient (5%) developed reactivation of varicella zoster treated with valacyclovir for 10 days, two patients (10%) developed increases in transaminases that self-resolved at week 12, and 16 patients (80%) developed creatine kinase levels at both week 4 and 12. The authors also stated that 12 out of 20 patients completed COVID-19 vaccination without stopping treatment; this finding is very important and is in accordance with the recommendations published in the literature regarding vaccination during biological or small molecule treatment [33,34,35,36].

None of these treatment-related AEs led to the discontinuation of treatment. In conclusion, the study conducted by the authors demonstrated excellent results in terms of the efficacy and safety of upatacitinib; it is the first real-life experience in the literature with a data at 24 weeks [33,34,35,36,37].

The limitation of this study remains the sampling of 20 patients, further studies will be needed to confirm the thesis proposed by the authors.

To date, regarding upadacitinib, an ongoing phase II trial (NCT04430855) with 68 patients is active, but not recruiting.

### 3.3. Tofacitinib

Tofacitinib is an inhibitor approved for the treatment of rheumatoid arthritis, psoriatic arthritis, and ulcerative colitis, but it has recently been used to treat alopecia areata [38,39,40].

Tofacitinib interferes with the JAK-STAT signaling pathway and suppresses the production of inflammatory mediators by inhibiting the enzymes JAK1 and JAK3 [38,39,40].

There are several studies in the literature that show how tofacitinib has demonstrated efficacy in other dermatological conditions such as atopic dermatitis [41,42,43] or systemic lupus erythematosus [44] or psoriasis [45].

As far as HS is concerned, to date, there are no clinical trials available in the literature. There is an ongoing trial concerning the treatment of patients with Down syndrome suffering from HS, which will end at the end of 2024 [46]. 

The only real-life article reported in the literature was described by Savage et al., who reported two cases of HS successfully treated with tofacitinib [47].

The first case regarded a 26-year-old man with a history of HS for about 10 years, had practiced therapies with infliximab and unspecified antibiotics with poor results; the authors reported the use of tofacitinib 5 mg/twice a day together with cyclosporine 5 mg/kg/die. At 8 months, cyclosporine therapy was tapered while tofacitinib treatment was discontinued at 12 months due to the excellent outcome both clinically and on the patient’s quality of life. No AEs were reported [47].

The second case regarded a 24-year-old woman with a history of HS for about 6 years. The patient had practiced infliximab therapy that caused worsening of the condition, and prior to initiating therapy with tofacitinib, she had practiced therapy with 1250 mg mycophenolate mofetil (MMF) along with courses of antibiotic therapy with amoxicillin clavulanicum, which resulted in slight improvements [47].

The authors reported using tofacitinib 5 mg/bdie combination therapy with MMF for about 3 years, where significant improvements were shown both clinically and in the patients’ quality of life. In 3 years, no AEs have been reported other than the development of shingles, which did not cause the discontinuation of the current therapy [47].

This article has two important limitations: on one hand, the too low number of patients involved, and on the other hand, both cases enjoyed combination therapy. In conclusion, however, the clinical data are promising, and certainly further studies will be needed to establish the efficacy and safety of tofacitinib in HS.

However, the literature also reports a case of paradoxical HS during tofacitinib therapy, as described by Shaharir et al. [48]. 

The authors reported the case of a 52-year-old woman with juvenile rheumatoid arthritis who after failure with etanercept had started therapy with tofacitinib (dosage not specified in the text). After 4 months of therapy, the patient had developed nodules and abscesses in the region of the right axillary cavity, which were treated surgically, so treatment with tofacitinib was discontinued and then resumed 6 weeks after surgery [48].

This is a very important issue for HS because paradoxical reaction is defined as the occurrence or exacerbation of a disease while under treatment either for the disease itself or for another disease [49,50]. In particular, the drugs that may more frequently cause HS paradoxical reactions are anti-TNF alpha drugs [49]. This finding is influenced by the wide use of these drugs for different conditions. However, they are the only biologic drug currently approved for HS, so it is necessary to find therapeutic alternatives for these patients who develop reactions from these drugs using other classes of biologic drugs [51,52,53].

### 3.4. Ongoing Clinical Trial 

Other studies have been conducted or are still ongoing on the use of JAK inhibitors in HS. Three kinase inhibitors are being tested in a phase II RCT (NCT04092452) with 192 patients. PF-06650833 is an Irak4 inhibitor, brepocitinib is a Tyk2/JAK1 inhibitor, and ropsacitinib is a Tyk2 inhibitor. There is also one phase I trial (NCT04772885) with 124 healthy participants testing KT-474, which is an Irak4 inhibitor [27].

## 4. Discussion

HS is a chronic inflammatory disease that occurs in regions where there is a greater presence of apocrine glands [54]. The primum movens of the pathology is the occlusion of the follicular ducts, leading to the formation of nodules, abscesses, and fistulous tracts, which may cause local superinfections [54].

The first line of treatment for mild HS involves the use of antibiotics, anti-inflammatory, corticosteroids, and hair laser epilation [55,56]. Moderate to severe forms may be eligible for biological therapies. 

Surgical treatment for HS is a valid alternative to the treatments mentioned; several cases have been reported in the literature that show good therapeutic results with this technique [57,58].

To date, thanks to recent studies, we are moving toward the role of cytokines in HS [58,59,60,61]; it has been clarified that there is an overexpression of IL-17A, IL-26, IFN-γ, IL-27, and IL-β and a concomitant downregulation of IL-22 in the lesions of HS patients [59,60]. Adalimumab, an anti-TNF agent, remains the only biologic drug approved for HS [61], with an efficacy rate reaching an approximately 60–70% mean efficacy rate in real-life [14].

During these years, there have been various studies where drugs targeting IL1, IL17, and IL23 cytokines have been tested [62]. The most promising results from real-life would seem to direct us toward the use of secukinumab, an anti-IL17A, which reports a mean efficacy rate of about 60% [14].

As stated above, pro-inflammatory cytokines are crucial for HS development so inhibition of the JAK/STAT pathway could help to regulate the expression of inflammatory factors such as IL-6 or IL-23 simultaneously [63,64,65,66,67].

Jak3 upregulation was found in the lesional tissue of HS patients [67]. The role of the JAK-STAT pathways in the regulation of cytokines, particularly type 1 cytokine receptors: ILs including IL-12 and IL-23 and type 2 cytokine receptors including IFN and IL-10-related cytokines [IL-10, IL-19, IL-20, IL-22, IL-26] could warrant a new therapeutic target for HS patients [67].

Unfortunately, there is no real resolutive therapy for HS due to the complexity of the pathogenesis, which is not yet well-defined; the efficacy results of the available treatments underline how necessary it is in the future to find new therapeutic targets.

In our review, we collected all of the data available to date on the use of JAK inhibitors in HS, showing promising results both in terms of the efficacy and safety; to date, it is certainly evident that high doses of JAK inhibitors provide good results, therefore, it will be essential in the future to investigate the safety of these high doses. We have also grouped together all of the clinical trials currently in progress, which give us hope for the future and which underline how the great attention paid by researchers toward this class of drugs.

The limit of our review was the too low sampling, but to date, no reviews dealing with this topic are available to the best of our knowledge.

## 5. Conclusions

In conclusion, in recent years, the interest in HS has strongly increased. To date, the only biological drug currently approved for HS is adalimumab. However, due to the chronic remitting course of the disease, there is still a huge unmet need for HS treatment.

Thanks to new studies, we are moving toward new classes of drugs against new immunological targets (IL-12, IL-17, IL-23, IL-36, CD-40, JAK family members, complement, LTA4 and CXCR1/2) that are under study [64,65,66]. 

There are still little data to understand the effectiveness of JAK inhibitors for HS. Particularly, there is only one published clinical trial in the literature (Janus kinase 1 inhibitor INCB054707), a real-life study with 15 patients up to week 24 with upadacitinib and a case series where tofacitinib was successfully used. Conversely, there are several ongoing clinical trials. The limited available data show promising results in terms of efficacy and safety. Hence, new studies are highly needed to test the validity of these new potential drugs for HS.

## Figures and Tables

**Figure 1 medicina-59-00801-f001:**
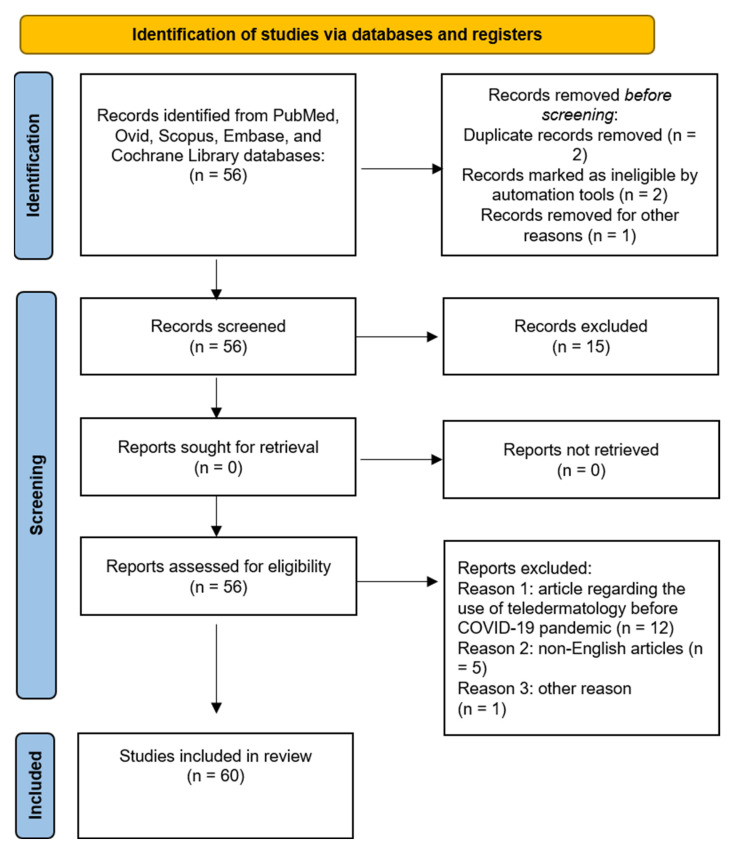
PRISMA checklist.

## Data Availability

The data are reported in the current study and are available upon request by the corresponding author.

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
