# Peer review of "Hidradenitis Suppurativa and JAK Inhibitors: A Review of the Published Literature"

_medicina, 2023, doi:10.3390/medicina59040801_

Round 1

Reviewer 1 Report

Abstract. correct "aprocrine"

Abstract. This sentence "Results to date in the literature show high levels of efficacy" seems to be too optimistic

Also in the intro please correct "aprocrine"

Introduction. I suggest to rephrase this sentence "mild and moderate forms are usually treated with long-term antibiotics with or without corticosteroid therapies, while severe forms are candidates for biological therapies..." as follows "mild HS forms are usually treated with conservative treatment such as topical resorcinol or clindamycin [Molinelli E, et al. Efficacy and safety of topical resorcinol 15% as long-term treatment of mild-to-moderate hidradenitis suppurativa: a valid alternative to clindamycin in the panorama of antibiotic resistance. Br J Dermatol. 2020; 183(6):1117-1119. doi: 10.1111/bjd.19337] or hair laser epilation [Nazzaro G, et al. High-frequency ultrasound in hidradenitis suppurativa as rationale for permanent hair laser removal. Skin Res Technol. 2019 Jul;25(4):587-588. doi: 10.1111/srt.12671 - Molinelli E, et al. Alexandrite laser as an adjuvant therapy in the management of mild to moderate hidradenitis suppurativa: A controlled prospective clinical study. J Am Acad Dermatol. 2022; 87(3):674-675. doi: 10.1016/j.jaad.2021.10.060.]; while moderate to severe forms undergo long-term antibiotics [van Straalen KR, et al. The efficacy and tolerability of tetracyclines and clindamycin plus rifampicin for the treatment of hidradenitis suppurativa: Results of a prospective European cohort study. J Am Acad Dermatol. 2021 Aug;85(2):369-378. doi: 10.1016/j.jaad.2020.12.089.] or may be candidates for biological therapies [Marzano AV, et al. Evidence for a 'window of opportunity' in hidradenitis suppurativa treated with adalimumab: a retrospective, real-life multicentre cohort study. Br J Dermatol. 2021 Jan;184(1):133-140. doi: 10.1111/bjd.18983].

In introduction you can add this sentence: "Skin ultrasound is currently used to diagnose HS [Nazzaro G, et al. Comparison of clinical and sonographic scores in a cohort of 140 patients with hidradenitis suppurativa from an Italian referral centre: a retrospective observational study. Eur J Dermatol. 2018 Dec 1;28(6):845-847] and moreover to evaluate clinical response to treatments as it demonstrated to be a valid non-invasive biomarker" [Nazzaro G, et al. Vascularization and fibrosis are important ultrasonographic tools for assessing response to adalimumab in hidradenitis suppurativa: Prospective study of 32 patients. Dermatol Ther. 2021 Jan;34(1):e14706. doi: 10.1111/dth.14706.] 

Please decide in the whole text if "phase 2" or "phase II"

Table 1 is missing

I do not have substantial concerns regarding methods and results of your study, which sounds scientifically and has been correctly set up.

In the Discussion this sentence "HS is a chronic inflammatory disease that occurs in regions where there is a greater presence of follicles" is not clear. According to this, the scalp should be involved more than axilla.

Discussion line 3: "leading to..." is correct

Discussione line 4: "which may cause local superinfections [Benzecry V, et al. Hidradenitis suppurativa/acne inversa: a prospective bacteriological study and review of the literature. G Ital Dermatol Venereol. 2020 Aug;155(4):459-463. doi: 10.23736/S0392-0488.18.05875-3.]

This sentence "The first line of treatment involves the use of antibiotics, anti-inflammatory, and corticosteroids" is too simplistic.

Author Response

Dear Reviewer, 

Thank you for your comments and suggestions.
We have modified the introduction as you requested and added the valuable references you have provided. Table 1 was a typo we edited.
We have corrected spelling errors.
We have edited the discussion part as you requested.
We thank you for the attention of your review

Best regards

Reviewer 2 Report

Dear Authors, 

Thank you for the Opportunity to read this manuscript. This

is meta-analysis due to the guidelines of PRISMA concerning JAK inhibitors in HS treatment. 

In my opinion you should make some corrections. 

my minor comments are below : 

1) Please elaborate on any abbreviations under figures and in the text

2) Introduction Section - in my opinion some information about the characteristics of the disease is missing i.e. there is no sentence about how the disease affects the quality of life - suggests PMID : 36686013 Yet still in the world the 1st line of treatment is antibiotic therapy , this results in increased multidrug resistance of bacteria that colonise HS lesions. PMID : 36686013 I think it is worth highlighting this given looking at the global problem of superbugs, this will also significantly increase the conclusions about biologic therapy. This will perhaps avoid in the future the overproduction of drug-resistant strains that are the result of antibiotics overtreatment in HS.

3) Introduction Section  or in Discussion- Biological treatment is described here, but there is at least one sentence missing about the surgical methods available for the treatment of HS - this is very important as in severe cases surgery gives very good results. I suggest that the authors include information on surgical treatment of HS in the introduction section. PMID : 36686007 PMID : 36004913

Congrats on an interesting paper which is very relevant. I think the authors will address any suggestions well.

Author Response

Dear Reviewer, 

Thanks for the comments and suggestions
We have checked all the text for the abbreviations and corrected them if necessary.
We put surgery in the discussion part as you requested and added part in the introduction as you suggested
We thank you for the comments.

Best regards

Round 2

Reviewer 1 Report

No further comments

Reviewer 2 Report

Accept as it is. Well adressed corrections